# DUPS: Dynamic upsampling for efficient semantic segmentation

## Abstract

We present **DUPS**, a coarse-to-fine vision transformer for semantic segmentation. Unlike models that begin with dense high-resolution tokens, DUPS starts at low resolution and dynamically upsamples only regions predicted to contain semantic boundaries, encouraging tokens to represent a single semantic class rather than a mixture of classes. Mixed-resolution attention enables interaction between coarse and fine tokens, allocating computation to semantically complex areas while avoiding redundant processing in homogeneous regions. Experiments on ADE20K, COCO-Stuff, and Cityscapes demonstrate that DUPS achieves state-of-the-art results on ADE20K and COCO-Stuff with substantially fewer FLOPs, and delivers competitive performance on Cityscapes at markedly lower compute. For example, DUPS-Base attains **54.6 mIoU** on ADE20K in the ∼110M-parameter class while using fewer FLOPs than comparable backbones.

## 1 Introduction

In semantic segmentation tasks, not all pixels carry equal importance. In a typical scene, large regions may consist of homogeneous areas such as sky, road, or walls, which contain little spatial or semantic variation. In such areas, dense per-pixel computation is unnecessary. Conversely, compact regions with high object density, fine structures, or class boundaries demand higher spatial resolution to ensure accurate predictions. This uneven distribution of semantic content motivates the need for architectures that dynamically adjust resolution based on local complexity.

Most computer vision architectures process images using a uniform grid of square patches, assigning equal computational cost to each patch regardless of its semantic content. For example, in Vision Transformers (Dosovitskiy et al., 2020), it is common to divide the input image into fixed-size patches (e.g., 16×16), each of which is embedded into a token, applying the same representation capacity to both simple and complex regions. Convolutional neural networks such as ConvNext (Woo et al., 2023) similarly apply filters over regularly spaced patches, implicitly treating all areas of the image with equal importance. While some recent models (Ziwen et al., 2023; Lu et al., 2023; Tang et al., 2023) introduce mechanisms for content-aware processing or adaptive computation, the majority of widely used architectures still rely on uniform spatial partitioning, which does not align with the uneven distribution of information in real-world images.

While uniform spatial partitioning is inefficient, most non-uniform approaches still follow a high-to-low resolution processing pipeline. In these architectures, the input image is initially represented at high resolution, and information is gradually compressed through pooling, striding, or token merging. As a result, all regions, regardless of their semantic importance, are initially processed at the finest spatial granularity. This leads to unnecessary computation in homogeneous or uninformative regions, and limits the potential efficiency gains of token pruning or merging in later stages. An ideal approach would avoid assigning high-resolution capacity to regions that do not require it in the first place, allocating computational resources only where semantic complexity demands it.

Beyond where computation is spent, there is also the question of how representational capacity is used within each token. When a token aggregates pixels from multiple semantic classes, its feature vector must encode the class mixture (which classes are present and in what proportions), the spatial layout and extent of each class within the patch, and the patch's overall position. By contrast, if a token predominantly corresponds to a single class, its representation can devote more dimensions to modeling intra-class variation and higher-order semantics rather than resolving class boundaries.

Constraining tokens to be largely class-specific improves the efficiency of the representation, allowing models with modest width to rival the semantic expressiveness of wider baselines.

To address both the computational and representational inefficiencies discussed above, we propose **DUPS** (Dynamic Upsampling with Mixed Resolution Tokens), a hierarchical encoder that processes images in a coarse-to-fine manner. Instead of starting with a dense high-resolution representation, DUPS begins with a low-resolution embedding and progressively upsamples tokens based on predicted semantic complexity. Using an predicted estimate of how strongly each token overlaps semantic boundaries, the model ranks tokens by how likely they are to mix classes and selectively refines those with the highest scores. In this way, mixed-class regions are split into several higher-resolution, more class-specific tokens, while clearly homogeneous areas remain compact. This ensures that no region is represented at a higher resolution than necessary. Tokens corresponding to semantically rich regions are selectively upsampled, while uniform areas remain compact. The architecture follows an inverted U-Net (Ronneberger et al., 2015) structure, expanding resolution toward the middle layers to capture fine details, and then reducing it again to refine higher order semantic structures. To summarize, our main contributions are:

- A boundary-aware scoring and selection module that upsamples regions that has semantic boundaries, with scale/sub-patch embeddings and lightweight image–feature fusion.

- A content-adaptive, per-scale ratio policy for training and inference that enables variable token budgets while remaining compatible with minibatch training.

- **DUPS:** A coarse-to-fine transformer that maintains multi-scale tokens and allocates capacity in deeper layers after high/low-resolution interactions, integrating the upsampling block and ratio policy into a single architecture.

- We demonstrate the effectiveness of DUPS on three challenging semantic segmentation benchmarks: ADE20K, COCO-Stuff, and Cityscapes, achieving state-of-the-art performance with significantly fewer FLOPs.

## 2 RELATED WORK

### 2.1 TOKEN DROPPING AND MERGING

Many methods improve transformer efficiency by removing tokens deemed uninformative or by aggregating nearby/similar tokens (Rao et al., 2021; Pan et al., 2021; Bolya et al., 2022; Marin et al., 2023; Long et al., 2023; Wei et al., 2023; Kim et al., 2024; Chang et al., 2023). These approaches typically start from a dense high-resolution grid of tokens and progressively reduce sequence length by pruning or merging similar tokens to gain efficiency. This yields a non-uniform, content-aware token density, but only through token removal: the spatial resolution never exceeds that of the initial grid, and training still processes an initially dense token set. In contrast, DUPS adopts the complementary strategy of starting from a coarse grid and splitting tokens only where additional detail is needed, increasing resolution locally while keeping homogeneous regions compact.

### 2.2 ADAPTIVE DOWNSAMPLING

A complementary direction makes tokenization or routing content-aware, e.g., saliency-driven multi-resolution grids, attention-guided sampling, or per-input policy decisions (Ronen et al., 2023; Fayyaz et al., 2022; Meng et al., 2022).

For dense prediction, AutoFocusFormer (AFF) (Ziwen et al., 2023) proposes point-based local attention with balanced clustering and a learnable neighborhood-merging module to support segmentation heads. Other works merge/share or prune tokens based on semantics or difficulty (Lu et al., 2023; Tang et al., 2023). While these approaches introduce adaptivity, most still start dense and then select, share, or prune tokens thereafter; they typically do not grow spatial resolution on demand during forward passes, and the initial training compute remains tied to a high-resolution tokenization.

## 2.3 ADAPTIVE UPSAMPLING

There is also a line of work on learned upsampling that takes context into account. Dynamic upsampling operators such as DySample (Liu et al., 2023) and frequency-aware fusion modules such as FreqFusion (Chen et al., 2024) act on dense feature maps: DySample learns sampling locations for each output position, and FreqFusion applies adaptive low-/high-pass filtering and offsets to boundary sharpness during upsampling. These methods focus on how to upsample (the upsampling operator itself), typically at all spatial locations, whereas DUPS focuses on what to upsample, starting from a coarse representation and increasing resolution only for selected tokens while keeping homogeneous regions compact. In this sense they are orthogonal and potentially complementary to our dynamic upsampling policy.

## 2.4 INVERTED STRUCTURES

Inverted pyramid designs redistribute capacity toward coarse scales or follow an overview-to-detail schedule (Zhu et al., 2024; Lou & Yu, 2025). OverLoCK (Lou & Yu, 2025) begins with a coarse global context derived from low-resolution features, and later refines these using fine-grained attention. These architectures alter how capacity is allocated across scales but generally keep resolution schedules fixed rather than content-driven within an image.

Beyond encoder design, boundary-refinement post-processing methods such as SegFix (Yuan et al., 2020) improve boundary quality by redirecting boundary pixels toward more reliable interior predictions using learned offsets. SegFix is model-agnostic and complementary to our approach; in principle it could also be applied on top of DUPS predictions, whereas DUPS incorporates boundary-awareness directly into the encoder.

Most prior work either compresses tokens after a dense beginning, modifies upsampling or fusion on dense feature maps, or redistributes capacity with fixed scale plans. A gap remains for architectures that begin coarse and allocate higher spatial resolution only where semantic complexity warrants it, while maintaining multi-resolution representations that interact across scales during dense prediction.

# 3 METHOD

DUPS is a hierarchical image encoder that creates multi-scale feature representations by dynamically upsampling lower-resolution tokens based on semantic importance. This allows us to focus computational resources on semantically complex areas. In particular, DUPS is designed so that tokens are encouraged to correspond to a single semantic class, rather than mixing multiple classes within the same token.

The architecture begins with a low-resolution representation of the input image, which provides computational efficiency by avoiding dense processing of uninformative regions. Many vision architectures apply token merging techniques after an initial high-resolution embedding. Instead DUPS begins at the coarsest level and progressively increases resolution only where needed. This allows the model to avoid ever allocating high-resolution tokens to semantically simple regions, enabling more principled and efficient use of computational resources.

As in most deep networks, deeper layers in DUPS encode increasingly abstract and high-level semantic features. These features typically require higher-dimensional embeddings to capture the necessary complexity. To maintain efficiency, the architecture therefore gradually reduces spatial resolution in later stages, using an inverted U-shaped pattern as seen in encoder-decoder architectures like U-Net. This design enables DUPS to preserve rich semantic representations and at the same time to keep the token count and memory footprint manageable.

## 3.1 ENCODER ARCHITECTURE

DUPS follows an inverted U-Net design, where patch resolution progressively increases and then decreases again, see Figure 1. To minimize the number of tokens, DUPS begins with a patch embedding layer with a $32 \times 32$ kernel to generate a compact set of low-resolution tokens. These are refined

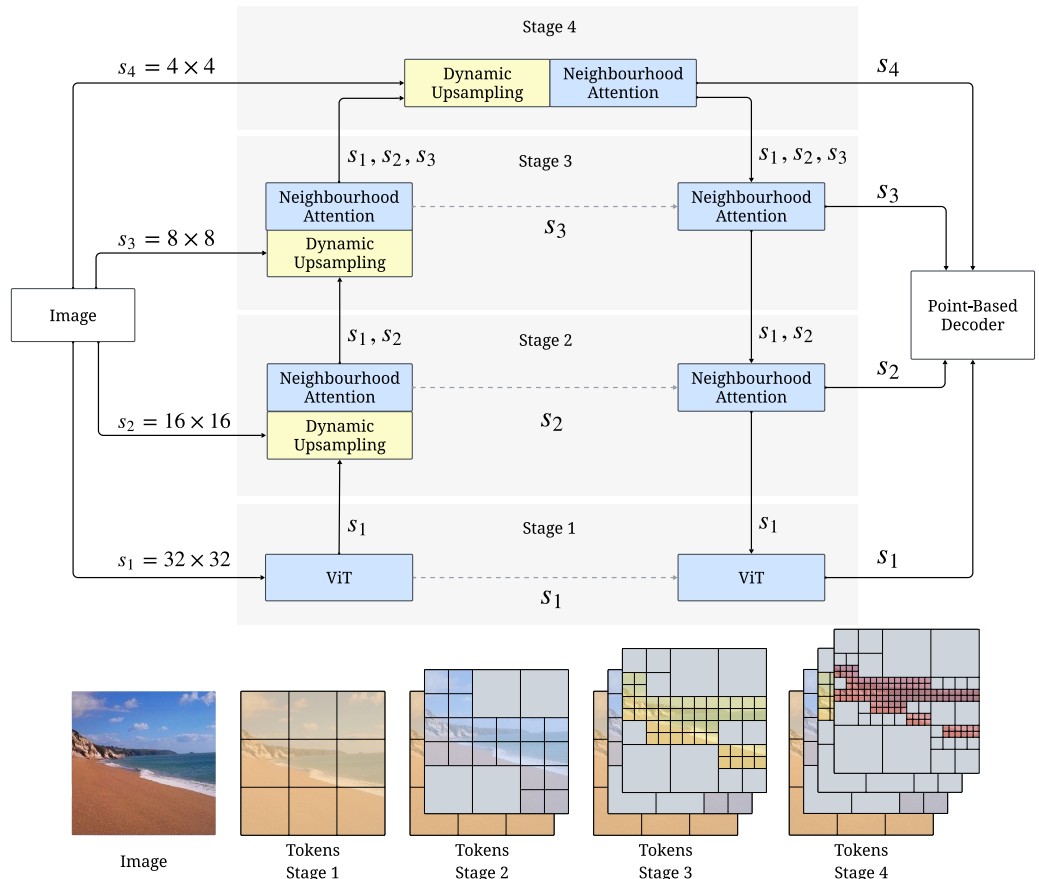

Figure 1: The DUPS architecture consists of an inverted U-Net encoder that produces sparse multi-scale tokens. The scales $s_1, s_2, s_3, s_4$ denote the patch size represented by each token. At stage $i$, the token set contains all tokens with scales $s_1, \ldots, s_i$, and Neighborhood Attention is applied jointly over this mixed-resolution set based on token positions. See Figure 2 for upsampling block details.

through a lightweight Vision Transformer and then passed into a dynamic upsampling block, where a subset of $K$ tokens is selected for spatial expansion based on predicted semantic importance.

The original low-resolution tokens and the newly upsampled high-resolution tokens are concatenated and passed into a neighborhood attention block from AFF (Ziwen et al., 2023). Neighborhood attention is both computationally efficient and crucial for handling sparse token sets, as attention windows are computed over dynamic, localized regions. There, attention is computed in regional clusters based on token locations, thus attention will be performed across resolution scales which enables interaction between coarse and fine representations given a mixed-resolution input. For example, small patches gain high-level context by attending to larger patches, and reciprocally larger patches are enriched with local detail as the upsampled tokens incorporate new image data. This process is repeated for every new, higher resolution scale, where only the highest resolution tokens are evaluated for additional upsampling.

After reaching the highest resolution, the encoder enters its descending path, following the right side of the inverted U-Net. At each layer in this phase, two parallel operations are performed. First, the currently highest-resolution tokens are forwarded to the multi-scale decoder. Second, the remaining lower-resolution tokens are projected to a higher-dimensional space and fused with residual tokens from the corresponding encoder layer on the left side of the U-Net. These combined tokens are then passed to the next layer, where they are further refined. This process is repeated at every stage, enabling coarser tokens to benefit from both higher representational capacity and interaction with fine-grained tokens, while preserving a compact token structure where possible.

Table 1: Architecture hyperparameters for three DUPS model scales. "Min. patch size" denotes the smallest image region represented by a token at that stage. "Dim" is the transformer channel width. $D_L$ and $D_R$ are the numbers of blocks on the left (ascending) and right (descending) sides of the U-Net, respectively.

| | | **DUPS-Tiny** | | | **DUPS-Small** | | | **DUPS-Base** | | |
|---|---|---|---|---|---|---|---|---|---|---|
| **Stage** | min patch-size | Dim | $D_L$ | $D_R$ | Dim | $D_L$ | $D_R$ | Dim | $D_L$ | $D_R$ |
| 1 | $32 \times 32$ | 512 | 1 | 4 | 512 | 2 | 3 | 768 | 2 | 4 |
| 2 | $16 \times 16$ | 256 | 1 | 16 | 256 | 2 | 24 | 384 | 2 | 18 |
| 3 | $8 \times 8$ | 128 | 1 | 4 | 128 | 2 | 6 | 192 | 2 | 6 |
| 4 | $4 \times 4$ | 64 | 4 | - | 64 | 2 | - | 96 | 8 | - |

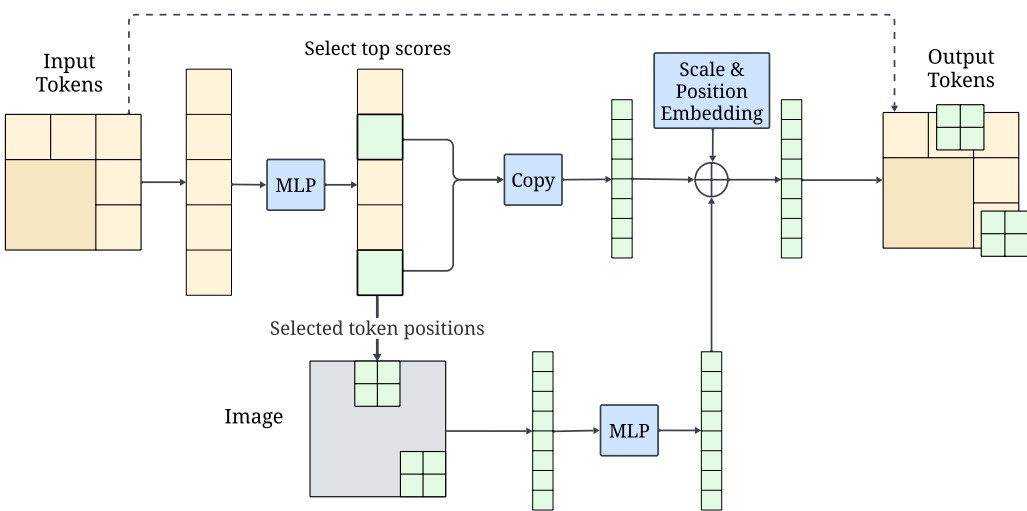

Figure 2: The dynamic upsampling block in DUPS with two different input scales (Stage 3). Upsampling scores are computed from the highest-resolution tokens; the top-scoring ones are cloned into 4 sub-tokens. Refined image tokens, scale embeddings, and relative position embeddings are added, and the new tokens are concatenated with the original tokens to form the final mixed-resolution token set.

We keep the left (ascending) path intentionally shallow to control early compute, while still performing cross-scale mixing at each stage. Greater capacity is assigned to the right (descending) path, where coarse tokens, having now attended to high-resolution tokens are further refined within a multi-resolution context. See Table 1 for configuration details.

## 3.2 DYNAMIC UPSAMPLING BLOCK

This block takes a set of tokens from the previous (lower) resolution scale and outputs an upsampling score $u_i$ for each token $i$. The score indicates the number of semantic edge pixels in the patch that the token represents. Then, the $K$ tokens with the highest scores are selected for upsampling where $K$ is computed from $K = \gamma \times N, 0 \leq \gamma \leq 1$, where $N$ is the number of input tokens and $\gamma$ is the dynamic upsampling ratio, see Section 3.2.1.

For each selected token, four copies are created for the $2 \times 2$ sub-patches. We experimented with interpolation and projection but did not observe performance gains, so we opted for repeated token copies to reduce parameter count. A learned scale embedding and a sub-patch position embedding is added to each copy. Finally, image data is extracted from the original image at the locations selected for upsampling. The new image data is refined through an MLP and then added to the upsampled tokens from the previous step. See Figure 2 for details.

### 3.2.1 UPSAMPLING SCORE

The Dynamic Upsampling block predicts an upsampling score per patch that represents the importance of upsampling that patch. The ground truth scores are generated by computing a binary map of semantic edge pixels from the segmentation ground truth, where a semantic edge pixel is a pixel that has a neighbor of a different class than itself, represented as a 1; if all neighbours are the same class, its represented as a 0. From the edge pixel map, we then sum all edge pixels in a patch, to create the ground truth upsampling score for that patch. The upsampling MLP is then trained to predict the normalized ground truth upsampling score using mean square error (MSE) loss.

The reasoning behind this representation is that we aim for each token to represent a single class, simplifying downstream reasoning for both encoder and decoder modules. Consider the alternative: a feature vector representing multiple classes must encode both the class identities and their spatial arrangements, increasing representational complexity. In comparison, if the feature vector only has a single class, the attention can solely be on the semantic meaning of this patch.

### 3.2.2 DYNAMIC UPSAMPLING RATIOS

Since each input image has a unique distribution of semantic content, the number of tokens that should be upsampled at a given scale should vary per sample. To accommodate this, we compute a dynamic upsampling ratio $\gamma_{i,s}$ for each token based on its predicted upsampling scores $u_i$ at scale $s$. Specifically, we define $\gamma_{i,s}$ as the fraction of tokens whose predicted score exceeds a threshold $\tau_s$:

$$\gamma_{i,s} = \frac{1}{N_s} \sum_{i=1}^{N_s} 1(u_i > \tau_s),$$ (1)

where $s$ is the current scale, $\gamma_{i,s}$ is the resulting upsampling ratio, $N_s$ is the number of input tokens, $u_i$ is the predicted upsampling score for token $i$, and $\tau_s$ is the threshold. The threshold $\tau_s$ is set to be slightly above zero, allowing for some tolerance in the predicted number of semantic edge pixels, and ensuring that only patches with sufficient predicted complexity are selected for upsampling.

This strategy works naturally at inference time when processing one sample at a time. However, during training with mini-batches, samples may yield different upsampling ratios, making it infeasible to apply per-sample dynamic token selection within a batch. To address this we compute the maximum batch upsampling ratio $\gamma_{\text{batch},s}$ across the batch and apply it uniformly:

$$\gamma_{\text{batch},s} = \min(\max(\max(\gamma_{0,s}, \gamma_{1,s}, \cdots, \gamma_{B,s}), 0.1), \beta_s)$$ (2)

where $B$ is the batch size and $\beta_s$ is a fixed ratio, pre-computed from the training set as the worst-case required fraction at scale $s$ plus an error margin, conditioned on the previous scale having been upsampled. The final $\gamma_{\text{batch},s}$ is used as the upsampling ratio for all samples in the batch. While this introduces a train–inference discrepancy in the upsampling ratios, it is a necessary compromise to enable batch-level training.

### 3.3 DECODER STRATEGY

Although large semantically homogeneous regions often are represented by a single token in DUPS, generating a full-resolution semantic segmentation map ultimately requires converting the sparse token representation back into a dense format. To accomplish this, we copy each coarse token across its corresponding spatial area and add a learned positional embedding for spatial differentiation.

We adopt the point-based Mask2Former (Cheng et al., 2022) decoder architecture introduced in AFF (Ziwen et al., 2023) which operates on multi-scale tokens. The decoder first applies a pixel decoder to process and align tokens across the four spatial scales. This is followed by a masked cross-attention module, which iteratively refines class queries by attending to the highest-resolution tokens in combination with the three lower-resolution token sets in alternating fashion.

In our implementation, we convert sparse tokens into a full-resolution token set after the pixel decoder but before the masked cross-attention. This design choice allows the decoder to benefit from computational efficiency of sparse representations during early processing while still enabling fine-grained correction and refinement through dense masked attention.

# 4 EXPERIMENTS

## 4.1 DATASETS

We evaluate our method on three publicly available semantic segmentation benchmarks: ADE20K (Zhou et al., 2019), Cityscapes (Cordts et al., 2016), and COCO-Stuff (Lin et al., 2014; Caesar et al., 2018). ADE20K is a scene parsing dataset containing 20,210 images annotated with 150 fine-grained semantic classes, covering a diverse range of indoor and outdoor environments. Cityscapes is a street-level driving dataset consisting of 5,000 high-resolution images with fine annotations for 19 semantic categories. COCO-Stuff extends the COCO dataset with dense pixel-level annotations, providing 172 semantic labels across 164,000 images.

## 4.2 EXPERIMENTAL SETUP

We build on Detectron2 with components from AFF and Mask2Former, and train on NVIDIA A100 GPUs. For complete details of the experimental protocol and additional settings, see Appendix A.

For pre-training, we use ImageNet classification. During this stage the upsampling blocks are disabled and tokens are upsampled at random according to a fixed ratio schedule (no content-based selection).

For fine-tuning, we follow AutoFocusFormer and Mask2Former for data augmentation and general hyperparameters. Models were optimized using AdamW with a learning rate of $4 \times 10^{-5}$. Crop sizes are fixed for all reported methods: 512×512 for ADE20K and COCO-Stuff, and 1024×1024 for Cityscapes. Training was conducted for 80k iterations with a batch-size of 32 on ADE20K and COCO-Stuff, and for 90k iterations on Cityscapes with a batch-size of 16. Exact values for $\tau_s$ and $\beta_s$ can be found in Appendix A.

For evaluation on ADE20K and COCO-Stuff, we resize the short side to 512 with preserved aspect ratio; for Cityscapes, we use overlapping 1024×1024 sliding-window inference. Performance is reported using mean Intersection over Union (mIoU)

FLOPs are measured for the full network at fixed input sizes: 512×512 (ADE20K/COCO-Stuff) and 1024×2048 (Cityscapes), reported as mean ± std per image over the validation set. For fairness, we report FLOPs only for methods evaluated at the same input resolution.

## 4.3 COMPARISON WITH STATE-OF-THE-ART METHODS

We provide baseline comparisons on three different benchmarks.

**ADE20K.** We compare **DUPS** at three model scales against state-of-the-art baselines with comparable parameter counts on the ADE20K validation set (Table 2). Across settings, DUPS attains higher mIoU while using fewer FLOPs. For example, DUPS-Tiny, show an mIoU improvement over efficient segmentation architectures such as AFF-Tiny-1/5 (+1.5), while simultaneously using lower FLOPs on average (-6.8G). The largest model we trained, DUPS-Base, also report the highest mIoU, an improvement of +0.9 compared to LRFormer-L with 55% less average FLOPs. DUPS also benefit from multi-scale testing, seeing large improvements (+1.1, +1.8, +1.1) to all model sizes, improving performance further.

**COCO-Stuff.** On COCO-Stuff, we observe the same pattern in Table 3 as on ADE20K: DUPS improves mIoU across model sizes while requiring substantially fewer FLOPs. In the ∼50M parameter regime, **DUPS-Tiny** attains 47.1/47.6, exceeding SegFormer-B3 and SegNeXt-L, while averaging only 41.7 GFLOPs per image, compared to 70–79 GFLOPs for baselines. At the largest scale, **DUPS-Base** delivers 48.5/49.6 mIoU, outperforming SegFormer-B5 and LRFormer-L with significant less compute (82.1 GFLOPs on average).

**Cityscapes** On Cityscapes, DUPS delivers competitive accuracy while using a fraction of the compute of comparable baselines (Table 3). Across model scales, DUPS attains mIoU close to state of the art, but with substantially lower FLOPs (e.g., **DUPS-Tiny** at 253.5 GFLOPs vs. 578–963 GFLOPs for similar-size baselines), providing a favorable accuracy–efficiency trade-off.

Table 2: Comparisons with state-of-the-art methods on ADE20K val.

| Backbone | Decoder | Params | FLOPs ↓ | mIoU (SS) ↑ | mIoU (MS) ↑ |
|---|---|---|---|---|---|
| ViT-CoMer-T (Xia et al., 2024) | UperNet | 38.7M | - | 43.0 | 44.3 |
| MiT-B3 (Xie et al., 2021) | SegFormer | 47.3M | 79G | 49.4 | 50.0 |
| MSCAN-L (Guo et al., 2022) | SegNeXt-L | 48.9M | 70G | 51.0 | 52.1 |
| SegMAN-B (Fu et al., 2025) | SegMAN | 51.8M | 58.1G | **52.6** | - |
| Swin-T (Cheng et al., 2022) | Mask2Former | 46.5M | 74G | 47.7 | 49.6 |
| AFF-Tiny-1/5 (Ziwen et al., 2023) | Mask2Former | 46.5M | 51G | 50.0 | - |
| DUPS-Tiny | Mask2Former | 48.5M | **44±7G** | 51.5 | **52.6** |
| ViT-CoMer-T (Xia et al., 2024) | UperNet | 61.4M | - | 46.5 | 47.7 |
| VMamba-T (Liu et al., 2024) | UperNet | 62.0M | - | 47.9 | 48.8 |
| ViT-Adapter-S (Chen et al., 2022) | UperNet | 57.6M | - | 46.2 | 47.1 |
| OverLoCK-Tiny (Lou & Yu, 2025) | UperNet | 63.0M | - | 50.3 | - |
| HRFormer-B (Yuan et al., 2021) | OCRNet | 56.2M | - | 48.7 | 50.0 |
| MiT-B4 (Xie et al., 2021) | SegFormer | 64.1M | 95.7G | 50.3 | 51.1 |
| LRFormer-B (Wu et al., 2025) | LRFormer | 69M | 75G | 51.0 | - |
| Swin-S (Cheng et al., 2022) | Mask2Former | 66.5M | 98G | 51.3 | 52.4 |
| AFF-Small-1/5 (Ziwen et al., 2023) | Mask2Former | 62.1M | 67.2G | 51.9 | - |
| DUPS-Small | Mask2Former | 61.7M | **60.7±9.3G** | **52.1** | **53.9** |
| ViT-CoMer-T (Xia et al., 2024) | UperNet | 144.7M | - | 48.8 | 49.4 |
| ViT-Adapter-B (Chen et al., 2022) | UperNet | 133.9M | - | 48.8 | 49.7 |
| VMamba-B (Liu et al., 2024) | UperNet | 122.0M | - | 51.0 | 51.6 |
| ConvNeXt V2-B Woo et al. (2023) | UperNet | 122.0M | - | 52.1 | - |
| OverLoCK-Base (Lou & Yu, 2025) | UperNet | 124M | - | 51.7 | - |
| MiT-B5 (Xie et al., 2021) | SegFormer | 84.7M | 183.3G | 51.0 | 51.8 |
| LRFormer-L (Wu et al., 2025) | LRFormer | 113M | 183G | 52.6 | - |
| SegMAN-L (Fu et al., 2025) | SegMAN | 92.4M | 97.1G | 53.2 | - |
| Swin-B (Cheng et al., 2022) | Mask2Former | 106.5M | 222.7G | 52.4 | 53.7 |
| DUPS-Base | Mask2Former | 111.5M | **82.4±14.0G** | **53.5** | **54.6** |

Table 3: Comparisons with state-of-the-art methods on the validation sets of COCO-Stuff and Cityscapes.

| Backbone | Decoder | Params | COCO-Stuff | | Cityscapes | |
|---|---|---|---|---|---|---|
| | | | Flops ↓ | mIoU ↑ (SS/MS) | Flops ↓ | mIoU ↑ (SS/MS) |
| MiT-B3 (Xie et al., 2021) | SegFormer | 47.3M | 79G | 45.5 / - | 963G | 81.7 / 83.3 |
| MSCAN-L (Guo et al., 2022) | SegNext-L | 48.9M | 70G | 46.5 / 47.2 | 578G | 83.2 / **83.9** |
| SegMAN-B (Fu et al., 2025) | SegMAN | 51.8M | 58.1G | **48.4** / - | 479G | **83.8** / - |
| Swin-T (Cheng et al., 2022) | Mask2Former | 46.5M | - | - / - | 537G | 82.1 / 83.0 |
| DUPS-Tiny | Mask2Former | 49.5M | **42±8G** | 47.1 / 47.6 | **254±18G** | 81.5 / 82.8 |
| HRFormer-B (Yuan et al., 2021) | OCRNet | 56.2M | 280G | 42.4 / 43.3 | 2224G | 81.9 / 82.6 |
| MiT-B4 (Xie et al., 2021) | SegFormer | 64.1M | 96G | 46.5 / - | 1241G | 82.3 / **83.9** |
| LRFormer-B (Wu et al., 2025) | LRFormer | 67M | 75G | 47.2 / - | 555G | **83.0** / - |
| Swin-S (Cheng et al., 2022) | Mask2Former | 66.5M | - | - / - | 732G | 82.6 / 83.6 |
| DUPS-Small | Mask2Former | 61.7M | **52±10G** | **47.4 / 48.2** | **356±27G** | 81.6 / 83.0 |
| MiT-B5 (Xie et al., 2021) | SegFormer | 84.7M | 112G | 46.7 / - | 1460G | 82.4 / 84.0 |
| LRFormer-L (Wu et al., 2025) | LRFormer | 111.0M | 122G | 47.9 / - | 908G | 83.2 / - |
| SegMAN-L (Fu et al., 2025) | SegMAN | 92.4M | 97G | **48.8** / - | 796G | **84.2** / - |
| Swin-B† (Cheng et al., 2022) | Mask2Former | 106.5M | - | - / - | 1050G | 83.3 / **84.5** |
| DUPS-Base | Mask2Former | 111.5M | **82±17G** | 48.5 / **49.6** | **584±38G** | 82.6 / 83.3 |

## 5 ABLATION STUDIES

**Dynamic Upsampling.** We ablate the effects of dynamic token selection and upsampling ratios at train and inference. When using dynamic ratios, we apply $\gamma_{i,s}$ at inference and $\gamma_{\text{batch},s}$ during training; for fixed ratios, we use the pre-computed $\beta_s$. The resulting budgets are $K_s = \lfloor N_s \gamma_{i,s} \rfloor$

(dynamic, infer), $K_s = \lfloor N_s \gamma_{\text{batch},s} \rfloor$ (dynamic, train), or $K_s = \lfloor N_s \beta_s \rfloor$ (fixed). In *full upsampling*, ratios are ignored. The results in Table 4 show that dynamic upsampling with dynamic ratios at both train and inference yields the best accuracy–efficiency trade-off, and even with the resulting train–inference mismatch in ratios it still outperforms both full upsampling and static-ratio baselines.

**Network architecture.** We ablate the rightmost descending path (U-Net) and the use of auxiliary image data. Three **DUPS-Tiny** variants are trained on ADE20K for 80k iterations from scratch (no pretraining). The baseline uses both the U-Net structure and auxiliary image data. The *No U-Net* variant keeps only the left (ascending) path, with parameters reallocated to match the baseline's count and distribution. As shown in Table 5, removing the descending path or the auxiliary image data both clearly reduces mIoU.

Table 4: Ablation of different upsampling strategies.

| Selection | Upsampling Ratio (Train) | Ratio (Inference) | FLOPs ↓ | mIoU ↑ |
|---|---|---|---|---|
| Full | - | - | 74.7G | 50.4 |
| Dynamic | $\beta_s$ | $\beta_s$ | 55.3G | 49.7 |
| Dynamic | $\gamma_{\text{batch},s}$ | $\beta_s$ | 55.3G | 50.3 |
| Dynamic | $\beta_s$ | $\gamma_{i,s}$ | 42.3±6.8G | 49.7 |
| Dynamic | $\gamma_{\text{batch},s}$ | $\gamma_{i,s}$ | 44.3±6.7G | 51.5 |

Table 5: Ablation of architecture choices on ADE20K. All models are trained from scratch (no pretraining); results are reported on the validation set.

| Method | mIoU ↑ |
|---|---|
| Baseline | 42.4 |
| No U-Net | 41.6 |
| No aux data | 41.0 |

**Training Memory Usage.** We measure training memory on Cityscapes with fixed $1024 \times 1024$ inputs on a single NVIDIA A100. AFF is the primary baseline, as it is a recent dynamic downsampling method in the same accuracy–efficiency regime. All models use pretrained initialization and identical settings. We report peak allocated GPU memory: after 20 warm-up steps, we record the per-step peak for 20 steps, resetting the counter each step and synchronizing. For each batch size we report the mean and maximum of these peaks. As shown in Table 6, DUPS requires substantially less memory across all batch sizes.

**Oracle Upsampling Scores.** We assess whether using ground-truth upsampling scores as an oracle during training improves segmentation. Models are fine-tuned on ADE20K; the upsampling blocks are always trained and used at inference with their predicted scores. An oracle rate of $x\%$ means we randomly use oracle scores for $x\%$ of training batches (and predicted scores otherwise). Results on validation set are shown in Table 7.

Training with oracle scores does not improve inference mIoU; higher oracle rates in fact degrade performance. During training we observe lower Dice and mask losses with oracle scores, but this reflects a train–test mismatch: the oracle effectively reveals boundary locations because upsampled tokens indicate class transitions, so the model learns to rely on this privileged signal rather than inferring boundaries from the image itself. At test time, when only predicted scores are available, this shortcut fails and accuracy degrades.

## 6 QUALITATIVE EXAMPLES

Figure 3 shows qualitative results across diverse scenes. For each input, we visualize the final prediction together with the tokens selected for upsampling at successive scales. The model consistently preserves semantically homogeneous regions (e.g., sky, road, walls) at coarse resolution, and upsamples along object boundaries and around fine structures and small instances. This behavior matches the design goal of allocating higher spatial capacity only where local semantic complexity is high.

## 7 CONCLUSION

We presented DUPS, a coarse-to-fine segmentation backbone that predicts semantic edge density early and upsamples only regions that warrant higher spatial detail. By beginning with

Table 6: Peak allocated training memory on Cityscapes with $1024 \times 1024$ inputs on a single NVIDIA A100.

|  | DUPS | | AFF | |
|---|---|---|---|---|
| **Batch-size** | Memory (Max/Mean) $\downarrow$ | | | |
| 1 | 6.2G | 5.2G | 8.6G | 8.6G |
| 2 | 10.8G | 9.5G | 14.6G | 14.6G |
| 4 | 20.3G | 18.2G | 27.5G | 27.4G |
| 8 | 38.7G | 35.7G | OOM | OOM |

Table 7: Oracle upsampling scores on ADE20K val. The "oracle rate" is the fraction of training batches using ground-truth scores.

| Model | Oracle rate | mIoU $\uparrow$ |
|---|---|---|
| DUPS-Tiny | 100% | 35.3 |
| DUPS-Tiny | 50% | 48.9 |
| DUPS-Tiny | 10% | 50.8 |
| DUPS-Tiny | 0% | **51.5** |

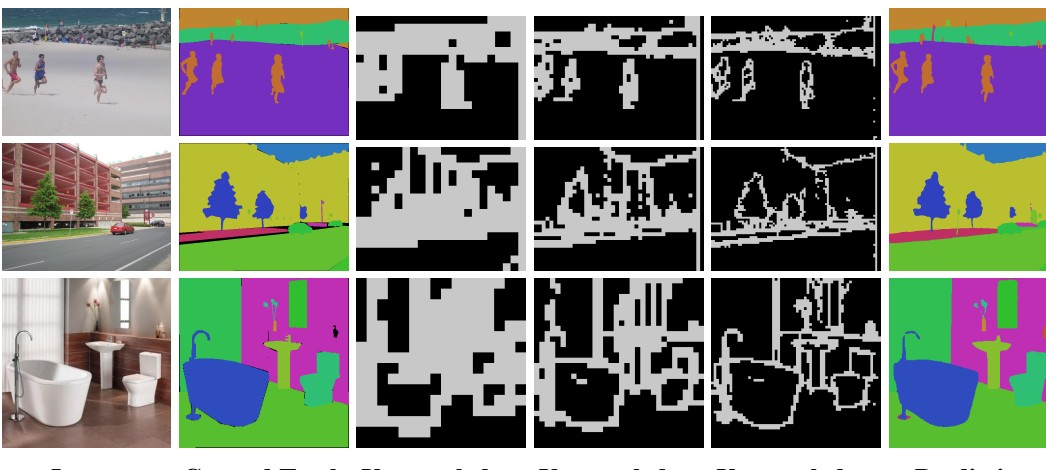

| **Image** | **Ground Truth** | **Upsampled $s_1$** | **Upsampled $s_2$** | **Upsampled $s_3$** | **Prediction** |

Figure 3: From left to right: original image, ground truth, $32 \times 32$ patches selected for upsampling, $16 \times 16$ patches selected for upsampling, $8 \times 8$ patches selected for upsampling, and prediction. Black in the ground truth means that the pixel was not labeled.

low-resolution tokens and dynamically increasing resolution where needed, DUPS avoids allocating high-resolution capacity to homogeneous areas and maintains multi-resolution representations that interact across scales. Experiments on ADE20K, COCO-Stuff, and Cityscapes show that DUPS achieves state-of-the-art accuracy on ADE20K and COCO-Stuff while substantially reducing FLOPs, and delivers competitive performance on Cityscapes at a fraction of the compute of comparable baselines. These results highlight that content-adaptive resolution control is an effective path to accurate and efficient dense prediction

A current limitation is that upsampling decisions are made only at the highest-resolution stage of each level; regions missed earlier cannot be revisited if scores are erroneous. Scalability to substantially larger backbones and longer pre-training also remains untested. Future work includes extending DUPS to panoptic, instance, and 3D segmentation, and enabling later stages to reconsider all candidate tokens for upsampling.

## REPRODUCIBILITY

Our Method section specifies the DUPS architecture and the dynamic upsampling block; Table 1 lists model-scale hyperparameters (patch-sizes/depths/dimensions). Section 4.2 provides key training settings, with additional details expanded in Appendix A. Experiments use public datasets (ADE20K, COCO-Stuff, Cityscapes) and dataset usage and splits follow standard practice. We include an anonymous, downloadable code package in the supplementary materials covering both pre-training and fine-tuning, along with configuration files for every model size and scripts to reproduce all benchmarks (ADE20K, COCO-Stuff, Cityscapes). Trained model weights will be released upon acceptance.

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

## A  ADDITIONAL EXPERIMENTAL DETAILS

**Hardware**  ImageNet pre-training was performed using 16 NVIDIA A100 GPUs; fine-tuning used 4 A100 GPUs, mixed-precision was enabled throughout.

**Pre-training**  As mentioned in Section 4.2 the model was pre-trained on the ImageNet classification task. During this phase, the dynamic upsampling blocks were disabled, and tokens were upsampled at random rather than based on predicted importance. A fixed upsampling ratio was applied uniformly across all spatial scales, starting at 100% (i.e., all tokens upsampled) and gradually decaying to 50% by the end of training. This strategy exposed the model to dense supervision early in training while encouraging sparser representations as learning progressed, facilitating better alignment with the dynamic token selection used during fine-tuning. For details regarding this choice, see Section C

**Fine-tuning**  Fine-tuning was performed on downstream semantic segmentation datasets using random resizing with a scale ratio between 0.5 and 2.0, random horizontal flipping, and random cropping on all datasets. Crop sizes were set to 512×512 for ADE20K, 1024×1024 for Cityscapes, and 512×512 for COCO-Stuff.

Optimization uses AdamW with backbone learning rate $4 \times 10^{-5}$, a $10\times$ multiplier for the decoder and upsampling blocks, and weight decay 0.05. We adopt a WarmupPolyLR schedule with 2500 warmup iterations (warmup factor 0.001).

For Cityscapes, we follow SegFormer and use three overlapping $1024 \times 1024$ windows for sliding-window inference. When multi-scale testing is used, we follow Mask2Former and resize the short side to $[256, 384, 512, 640, 768, 896]$ for ADE20K/COCO-Stuff and $[512, 768, 1024, 1280, 1536, 1792]$ for Cityscapes (with aspect ratio preserved) and include horizontal flips at each scale; predictions are aggregated to produce the final result.

Table 8 lists the upsampling hyperparameters. The thresholds $\tau_s$ are applied to predicted edge-pixel fractions per patch: e.g., for a $16 \times 16$ patch and $\tau_2 = 0.02$, the patch is upsampled if the predicted edge count exceeds $256 \times 0.02 \approx 5.1$ pixels. These values were set based on observed prediction noise on the training set.

The $\beta_s$ values are derived from ground-truth segmentation on each training set. At scale $s$, a patch is marked as "requires upsampling" if its GT map contains more than one edge pixel; per image, we compute the fraction of such patches. We set $\beta_s$ to the 99th-percentile of these per-image fractions and add a 0.05 safety margin.

Table 8: Upsampling Hyperparameters

| Parameter | CityScapes | ADE20K | COCO-Stuff |
|---|---|---|---|
| $\tau_1$ | 0.005 | 0.01 | 0.01 |
| $\tau_2$ | 0.01 | 0.02 | 0.02 |
| $\tau_3$ | 0.02 | 0.04 | 0.04 |
| $\beta_1$ | 0.5 | 0.85 | 0.7 |
| $\beta_2$ | 0.7 | 0.7 | 0.7 |
| $\beta_3$ | 0.65 | 0.6 | 0.6 |

**Upsampling block MLP sizes** The upsampling block uses a 3-layer MLP to predict the upsampling score $u_i$; its input and hidden widths equal the token dimension of the preceding layer.

For image data, raw pixels from the selected patch are first projected with a linear layer of size $3 \times (\text{patch-size})^2 \to d$, where $d$ is the preceding layer's token dimension. The projected vector is then refined by a 2-layer MLP with input/hidden/output width $d$.

## B  ADDITIONAL QUALITATIVE RESULTS

In Figure 4, additional qualitative results can be found

## C  IMAGENET PRE-TRAINING.

We investigate the effects of pre-training with random upsampling versus full upsampling and evaluate the final results after finetuning on ADE20K and Cityscapes. In the results in Table 9 we can see that even if the model using full upsampling performed better on imagenet (which is to be expected), by forcing the model to handle sparse tokens, downstream performance was improved.

We compare ImageNet pre-training with full upsampling versus random upsampling with upsampling blocks disabled and fixed ratio schedule as described in Appendix A. We then evaluate downstream performance after fine-tuning on ADE20K and Cityscapes. As shown in Table 9, the full-upsampling model attains higher ImageNet accuracy (as expected since upsampling performed randomly), but the random-upsampling model yields better segmentation mIoU. This suggests that exposing the backbone to sparse representations during pre-training improves its robustness to mixed-resolution inputs used at fine-tuning and inference.

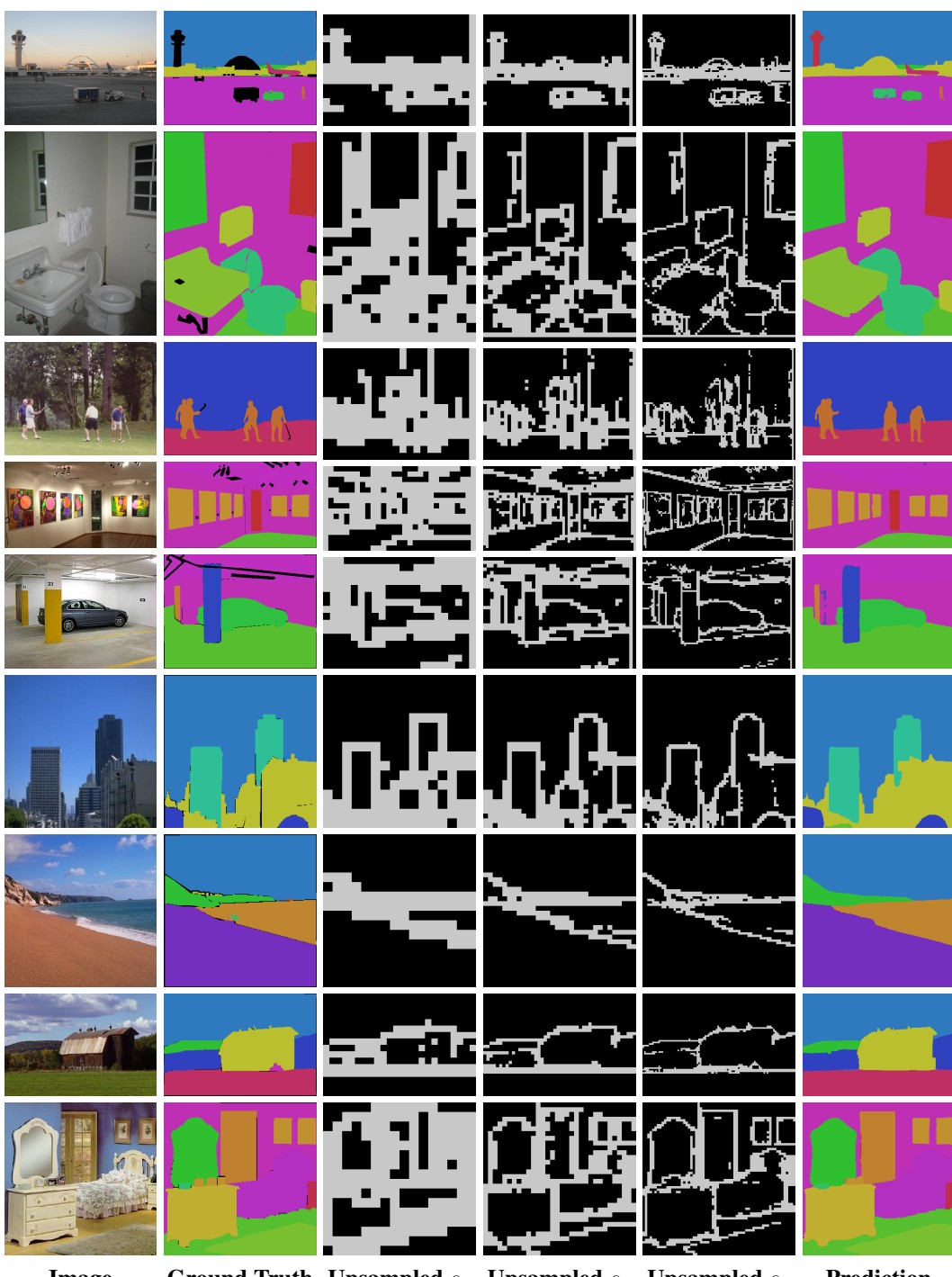

| **Image** | **Ground Truth** | **Upsampled** $s_1$ | **Upsampled** $s_2$ | **Upsampled** $s_3$ | **Prediction** |

Figure 4: From left to right: original image, ground truth, $32 \times 32$ patches selected for upsampling, $16 \times 16$ patches selected for upsampling, $8 \times 8$ patches selected for upsampling, and prediction. Black in the ground truth means that the pixel was not labeled.

## D  UPSAMPLING SCORE LOSS ABLATION.

We investigate the effect of using mean-squared error (MSE) versus mean absolute error (MAE) as the loss function for the upsampling score prediction blocks. The model used is DUPS-Tiny,

Table 9: ImageNet pre-training ablation: full vs. random upsampling. Top-1 accuracy is reported on ImageNet; mIoU is evaluated after fine-tuning.

| Method | Up-ratio | ImageNet Acc1 | ADE20K mIoU | Cityscapes mIoU |
|---|---|---|---|---|
| DUPS-Tiny | 100% | 81.2 | 50.8 | 80.6 |
| DUPS-Tiny | 50% | 78.9 | 51.5 | 81.5 |

pretrained on ADE20K for 80K iterations. As shown in Table 10, supervising the upsampling score with MSE yields clearly better performance than MAE. This is intuitive: small deviations in the predicted score are often negligible, as they typically do not change whether a token is selected for upsampling, whereas larger errors can flip the upsampling decision. Squaring the error places more emphasis on these larger mistakes, which leads to improved mIoU.

Table 10: Ablation study on the effect of loss function on the upsampling score

| Method | Loss Function | mIoU |
|---|---|---|
| DUPS-Tiny | MAE | 49.2 |
| DUPS-Tiny | MSE | 51.5 |