# OpenReview forum: "DUPS: Dynamic upsampling for efficient semantic segmentation"
_ICLR.cc/2026/Conference — Submitted to ICLR 2026_

### Official Review · Reviewer_2E8y · 2025-10-25

**Soundness:** 3
**Presentation:** 2
**Contribution:** 2
**Rating:** 4
**Confidence:** 5

**Summary:**

This paper introduces DUPS, a coarse-to-fine vision transformer for semantic segmentation. DUPS employs an inverted U-Net design, which selectively allocates computation to semantically complex areas, thereby reducing computational cost. The method is evaluated on ADE20K, COCO-Stuff, and Cityscapes, demonstrating promising performance with reduced computational cost.

**Strengths:**

- The dynamic upsampling design is an interesting approach, utilizing a learned predictor to estimate semantic edge density and focus computational resources on boundaries and fine structures.
- The proposed inverted U-Net architecture is a promising design choice for refining tokens while preserving multi-scale features.
- Experiments on different datasets show good performance.

**Weaknesses:**

- The proposed upsampling strategy is not convincing enough. Specifically, the authors fail to provide detailed proof of the benefits of the inverted U-Net design in upsampling from low to high resolution. Moreover, the inverted U-Net architecture bears resemblance to conventional encoder-decoder networks, where inputs are fed into a ViT and progressively upsampled while combining multi-scale features. This raises questions about the originality of the proposed architecture.

- A significant concern is the discrepancy between training and inference in the dynamic upsampling mechanism. During training, a batch-wise upsampling ratio is used, whereas inference employs a sample-specific ratio. Although the use of a pre-computed ratio is a necessary engineering solution, the paper neglects to analyze the potential impact of this mismatch.

- The upsampling process appears irreversible, with decisions not to upsample a region at a coarse scale being final. This design choice introduces a risk of compounding errors, where the upsampling predictor's failure to detect thin structures or small objects at an early stage cannot be recovered later.

- The paper should provide latency comparisons for different methods.

- Is the proposed Dynamic Upsampling a generic method? The paper lacks experiments demonstrating the effectiveness of Dynamic Upsampling on different feature backbones.

- The paper is difficult to follow due to the lack of clear and concise visualizations. The authors rely heavily on textual descriptions to explain the missing information in the figures, making it challenging for readers to understand the proposed method.

- Comparisons with some recent methods for semantic segmentation, such as SegMAN (CVPR'25) and FreqFusion (TPAMI'24), are missing. Note that I will not dismiss the contributions of this paper solely because its performance is not on par with SOTA methods. However, the authors should provide in-depth insights into how the proposed method can benefit the community compared to these SOTA methods.

**Questions:**

Please address the weakness section.

---

> ### Author Response · Authors · 2025-11-21
> **Answer to reviewer 2E8y**
>
> We thank the reviewer for the constructive feedback. Below we address the concerns raised in the weakness section.
>
> ## Inverted U-Net.
>
> We agree that encoder–decoder structures and U-Net–style multi-scale fusion are well established, and we do not claim these aspects as novel. Our architectural contribution is to invert the usual resolution schedule and couple it with dynamic sparse mixed-resolution tokens.
>
> Most segmentation backbones start from a dense high-resolution grid and progressively reduce spatial resolution before upsampling in the decoder. In contrast, DUPS starts from a coarse token grid, dynamically increases resolution, then reduces it again in the second half of the network. This inverted U-shaped path is not standard: early layers operate on very few tokens, and high-resolution capacity is used only near boundaries and fine structures.
>
> Regarding the concern that we “fail to provide detailed proof” of the benefits of the inverted U-Net, Table~5 explicitly ablates this design: we compare with and without the inverted U-shaped path and observe that adding the high-to-low leg on top of the coarse-to-fine path improves mIoU.
>
> ## Train–inference discrepancy.
>
> We acknowledge the discrepancy between the training and inference upsampling policies. Table~4 explores this by comparing fixed and dynamic upsampling settings, including cases where train-time and test-time ratios differ. The results show that, despite this mismatch, dynamic upsampling consistently outperforms fixed-ratio and full-upsampling baselines. In the revised manuscript, we clarify this discrepancy and emphasize that the batch-wise ratio is an engineering compromise that still improves over fixed policies.
>
> ## Upsampling irreversibility.
>
> We agree that the irreversibility of the upsampling decisions is a real limitation, and this was already acknowledged in the conclusion of the original submission. Even with dense edge supervision, the predictor can miss thin structures or small objects, and such regions may never be refined. Coarse tokens can still carry information about underlying fine structures, which the decoder can sometimes sharpen, but some failures are unrecoverable if a region is never upsampled. Allowing arbitrary “late” upsampling at all stages is non-trivial: each stage would require separate upsampling blocks for each input scale, and blocks for rarely changing decisions would receive little or no gradient. Given our focus on a lightweight design, we leave more flexible multi-stage refinement or top-down correction mechanisms as promising future work.
>
> ## FPS.
> In the revised version, we will add end-to-end inference FPS for DUPS.
>
> ## Generality.
>
> The reviewer asks whether Dynamic Upsampling is a generic method that can be demonstrated on different feature backbones. Conceptually, using a learned score to decide which regions to refine is generic and could be combined with any backbone that supports sparse and mixed-resolution features. In practice, our implementation uses a transformer backbone with Neighborhood Attention, which naturally supports local attention over such sparse token sets. Backbones that assume dense feature grids and fixed local neighborhoods (standard CNNs, fixed-window attention) are not directly compatible, and a pure ViT becomes prohibitively expensive at high resolutions. Because few existing backbones support local attention over sparse, mixed-resolution tokens, we restrict our experiments to the presented encoder and leave extensions to other architectures as future work.
>
> ## Visual clarity.
>
> In the revised manuscript, we improve the visual design of Figures 1 and 2, add a visual example to Figure 1 showing how the sparse token set evolves across stages, and refine the captions to reduce reliance on long textual explanations. We hope these changes make the overall architecture and dynamic upsampling process easier to follow.
>
> ## Baselines.
>
> SegMAN was mentioned by another reviewer and is an important baseline. We have updated our experimental tables to include it. While SegMAN achieves strong results using local attention and state-space models, DUPS explores a different design axis: we start from coarse tokens and dynamically refine only where needed, maintaining a sparse mixed-resolution representation with substantially reduced computation. In principle, state-space modules similar to those used in SegMAN could also be integrated into our mixed-resolution backbone, so we view these directions as complementary. Regarding FreqFusion, this method, like DySample and related works, performs content-adaptive upsampling at all spatial locations, whereas DUPS is dynamic in which tokens are upsampled, maintaining a sparse mixed-resolution representation. FreqFusion also targets different parameter regimes and training setups, making direct numerical comparison less straightforward. In the revised manuscript, we add FreqFusion to the related work and briefly discuss these differences.

---

> > ### Comment · Reviewer_2E8y · 2025-11-22
> >
> > Thank you for your response. However, my concerns are not adequately addressed. I would like to reiterate and elaborate on the issues that need to be addressed.
> >
> > The authors mentioned that most segmentation backbones start from a dense high-resolution grid and progressively reduce spatial resolution before upsampling in the decoder. I think that the proposed architecture is actually similar, where the original high-resolution input is downsampled to a lower resolution using a ViT encoder and then gradually upsampled. While incorporating multi-scale context from the original input during upsampling is an interesting idea, the overall architecture's differences from conventional segmentation architectures are overclaimed.
> >
> > Regarding Table 5, I understand the authors' motivation, but simply removing modules does not accurately reflect the effectiveness of the proposed method. The reduced model complexity naturally leads to decreased performance, which is intuitive. The authors should replace the removed modules with alternative methods to provide a more comprehensive and fair evaluation.
> >
> > I still have concerns about the upsampling ratio. Could you provide experiments with different ratios on various resolutions, such as 640x640 and 512x1024?
> >
> > The authors should evaluate the contribution of Natten to the performance and replace it with other modules, including removing the module, on different datasets. This is essential, as there are numerous local mixers available.
> >
> > The generalizability of the proposed Dynamic Upsampling is unclear, and it is uncertain whether this module can be widely adopted by the community. Furthermore, a fair comparison with other advanced upsampling methods is lacking, making it unclear what advantages this method offers. These issues fall within the scope of this paper and should be addressed rather than relegated to future work.
> >
> > FPS is an important efficiency metric in semantic segmentation, as FLOPs only reflect theoretical efficiency. Considering the performance gap with state-of-the-art segmentation networks, including SegMan, and the missing efficiency evaluation, the advantages of the proposed method are unclear.
> >
> > Overall, the paper requires heavy revisions to address the issues raised and is not acceptable in its current shape. I have also noticed that other reviewers have similar concerns.

---

### Official Review · Reviewer_XfA3 · 2025-10-28

**Soundness:** 2
**Presentation:** 1
**Contribution:** 2
**Rating:** 2
**Confidence:** 5

**Summary:**

This work studies the efficiency–accuracy trade-off in semantic segmentation. Motivated by the uneven distribution of semantic content, the authors propose DUPS, an inverted U-Net architecture. In DUPS, semantically simple regions are represented by a single token, while semantically complex regions are represented by multiple tokens. Specifically, DUPS incorporates a shallow, coarse-to-fine encoder and a relatively deep decoder network. Quantitative and qualitative experiments demonstrate the effectiveness.

**Strengths:**

- The method is grounded in the well-recognized observation that content in natural images is unevenly distributed, motivating the straightforward solution.
- The design of DUPS is concise and easy to grasp.

**Weaknesses:**

**Limited novelty**:
- From Table 1, we see that the right branch is much deeper than the left. The right branch is essentially a conventional neural network, while the left branch is a shallow pathway that directly injects image data to preserve spatial details. This dual-path design has been extensively studied in the semantic segmentation community over the past decade.
- The overall design is also very similar to a class of "learning-to-downsample" methods that allocate fewer tokens to object interiors and more tokens near object edges. I cannot recall the exact citation, but there is a series of works following this approach.

**Poor presentation**:
- DUPS can be viewed as the inverse of token-merging methods, which merge semantically similar tokens. Both approaches are based on the same assumption of uneven image-content distribution. However, the paper lacks any discussion of these token-merging methods, neither textual analysis nor quantitative comparison.
- DUPS selects patches for upsampling when their upsampling scores exceed a threshold. Therefore, introducing the notion of the *dynamic upsampling ratio* is redundant. Defining \(K\) as an indicator function (e.g. $\(K=\mathbf{1}(\{u>\tau\}))$) is sufficient and more concise.

**Questions:**

- Line 193, what is each feature $i$? It is suggested to use consistent terminology. Refer either to "each token" or to "each patch" throughout the manuscript.
- The effectiveness of the MSE loss is unclear because no ablation study is provided.

---

> ### Author Response · Authors · 2025-11-21
> **Answer to reviewer XfA3**
>
> We thank the reviewer for the feedback. Below we address each concern and clarify the design and positioning of DUPS.
>
> ## Dual-path design.
>
> The reviewer characterizes Table~1 as a deep “right branch’’ and a shallow “left branch’’ and concludes that such dual-path designs are well studied. We believe this characterization is too coarse and largely ignores what is specific to DUPS. Framing the architecture only as a “dual-path’’ network could apply to almost any model that reuse image features, and does not engage with the dynamic, sparse and mixed-resolution behavior that is central to our design. At a high level, the network does have two paths, and we do not claim this layout as novel; our contribution lies in how they are used and in the inverted resolution schedule.
>
> The “right branch’’ is not a standard dense encoder: it operates on a sparse, mixed-resolution token set produced by dynamic upsampling and applies neighborhood attention over multiple resolution scales. The “left branch’’ is not just an image-injection shortcut, but the pathway where we start from a coarse representation and perform boundary-guided sparsification via the upsampling scores, deciding which regions are refined and which remain compact.
>
> ## Learning-to-downsample methods.
>
> The reviewer notes similarities to learned downsampling methods. We already discuss this family of models in Sec.~2.2 (Adaptive downsampling), where we review content-aware tokenization and routing methods.
>
> We agree that this is a related direction, but learning to downsample a dense high-resolution grid and learning where to upsample from a coarse grid are fundamentally different formulations. Both lines of work share the observation that semantic content is unevenly distributed and that more capacity should be spent near boundaries and complex regions. However, most learned downsampling approaches begin with a dense tokenization and then decide how to reduce or redistribute resolution, whereas DUPS starts coarse and performs dynamic upsampling, increasing resolution only where needed. Without specific citations it is hard to match the exact works the reviewer had in mind, but we believe Sec.~2.2 already acknowledges and positions DUPS relative to this class of methods.
>
> ## Token-merging methods.
>
> The reviewer suggests that DUPS can be viewed as the inverse of token-merging methods and states that the paper ``lacks any discussion'' of these approaches. We respectfully disagree with this characterization. Sec.~2.1, titled “Token dropping and merging’’, already discusses token-pruning and token-merging methods that remove or merge tokens based on content.
>
> At a high level, we agree that this line of work is related and that token-merging can be seen as complementary to our approach: most token-merging methods start from a dense high-resolution grid and progressively merge or drop similar tokens to gain efficiency, whereas DUPS starts from a coarse grid and upsamples tokens only where additional detail is needed. However, this is not a strict ``inverse'' relationship, since DUPS never reconstructs a full dense feature map in the encoder but instead maintains a sparse mixed-resolution token set throughout.
>
> Quantitative comparison is challenging since most token-merging methods are evaluated for classification only. The one segmentation-oriented example we are aware of, ``Content-aware Token Sharing for Efficient Semantic Segmentation with Vision Transformers'', reports results either on different datasets or on ADE20K using much larger models with ImageNet-22k pre-training, and is therefore not directly comparable to our setting.
>
>
> ## Dynamic ratio vs. indicator.
>
> At the token level, we agree that selection can be written as an indicator, e.g. $K = \mathbf{1}(u > \tau)$. However, our method also needs to control selection at the batch level: we must decide how many tokens to refine per image and per scale when different images have different numbers of tokens above threshold. For this purpose, an aggregate quantity such as the dynamic upsampling ratio $\gamma$ is needed, since it summarizes the indicator decisions as a fraction that can be compared and bounded across images in a mini-batch. In this sense, the ratio is not redundant with the indicator: the indicator expresses per-token decisions, while $\gamma_{\text{batch},s}$ provides the batch-level quantity we actually use to implement dynamic upsampling during training.
>
>
> ## Terminology.
> In this instance, we refer to a token. We agree the terminology should be more consistent and have revised the text to use “token’’ for elements of the mixed-resolution feature representation and “patch’’ only for the corresponding image region.
>
> ## MSE loss.
> In the revised version, we have added an ablation where we replace the MSE loss with an MAE loss for supervising the upsampling score.

---

> > ### Comment · Reviewer_XfA3 · 2025-11-28
> >
> > I have carefully reviewed the authors’ response and the comments from the other reviewers. I concur with Reviewer 2E8y that the architectural contribution may be overstated. According to Table 1, the conventional high-to-low pathway is significantly deeper than the left pathway, which runs counter to the coarse-to-fine narrative proposed in the paper. Rather than relying on the claimed architectural novelty, a more convincing evaluation would be to apply the proposed upsampling method to well-established frameworks such as FPN or U-Net and demonstrate consistent improvements there.

---

### Official Review · Reviewer_jsQa · 2025-10-29

**Soundness:** 3
**Presentation:** 2
**Contribution:** 2
**Rating:** 4
**Confidence:** 4

**Summary:**

The paper presents DUPS, a backbone designed for semantic segmentation that use a coarse-to-fine vision transformer that begins with low-resolution tokens and dynamically upsamples only regions predicted to contain semantic boundaries. The dynamic upsampling block predicts per-patch edge-density scores to select tokens for 2×2 sub-patch expansion and fuses lightweight image features with learned scale/sub-patch embeddings. Mixed-resolution neighborhood attention enables interaction between coarse and fine tokens. The decoder adapts Mask2Former/AFF. The paper shows promising performance compared to baselines, but crucial experiments are missing and the comparison is not fair among backbone baselines.

**Strengths:**

- The selective upsampling and mixed-resolution neighborhood attention is efficient and allows explicit cross-scale interactions during attention. The inverted U-Net for coarse-to-fine resembles the human visual system.
- Results demonstrate promising performance across three standard benchmarks under comparable number of parameters and FLOPs.

**Weaknesses:**

Missing experiments:
- Since the main contribution of this paper is the encoder, it should be comared fairly with other encoders to isolate the performance gains introduced by the novel encoder. Currently, the baselines use different decoders (e.g., OverLoCK use UperNet) and I cannot see whether the performance gains of DUPS are due to the novel encoder or the Mask2Former decoder. The authors should report segmentation results of different encoders using the same decoder in Table 2 and Table 3.
- The upsampling approach proposed should be compared with previous dynamic upsampling methods, such as DySample.
- Missing baselines and comparison: SegNeXt, SegMAN, EDAFormer (these models' decoders are a part of their contribution so comparing them directly with different decoders is fine). Also since DUPS uses edge map to train the dynamic upsampling block, it should be comapred with other methods that use edge maps such as SegFix.
- Latency/FPS of DUPS should be reported.
-  The proposed encoder performance on ImageNet-1k should be reported and compared to the encoders of other baselines.

Figures:
- The overall figures do not look very professional and should be polished (Figure 1, 2). Details of the upsampling block should be shown in the figure.

References

Learning to Upsample by Learning to Sample. ICCV 2023

**Questions:**

- I am curious on how many tokens are upsampled on each stage on average? Is it a sparse set of tokens?
- Can the method be extended to other dense prediction tasks?

Overall I like the idea of this work but I want to see the impact of the proposed encoder on mIoU when compared to other encoders with similar complexity and using the same decoder.

---

> ### Author Response · Authors · 2025-11-21
> **Answer to reviewer jsQa**
>
> We thank the reviewer for the detailed and constructive feedback. Below we address the main concerns and clarify the design and evaluation of DUPS.
>
> ## Results should clearly show the decoder used.
> We agree that the encoder contribution should be evaluated as fairly as possible. In our experiments we already report results for several encoders with a shared Mask2Former decoder (e.g., Swin and AutoFocusFormer). In the revised version, we make this more explicit by adding a separate column in Tables 2 and 3 indicating the decoder used, and clearly grouping the rows that use Mask2Former, so that the effect of changing the encoder only is easier to see.
>
> ## Comparison with DySample.
>
> We thank the reviewer for pointing out DySample. DySample and DUPS address complementary aspects of upsampling: DySample focuses on how to upsample a dense feature map, whereas DUPS focuses on which regions should be represented at higher resolution. We have added DySample to the related work section, clarifying this distinction and positioning DUPS as orthogonal and potentially complementary. A hybrid design that uses a DySample-like operator inside our Upsampling block is interesting future work but beyond the scope of this paper.
>
>
> ## Missing baselines.
> SegNeXt-L is already included as a backbone in our main comparison tables and other versions of SegNeXt does not match the parameter classes that we compare against.
> We thank the reviewer for pointing out SegMAN, this baseline was indeed missing and we have updated the tables to include SegMAN in our comparisons.
> For EDAFormer, reported results are all outside the parameter classes that we focus on and is therefore not included for a fair comparison.
> SegFix is a learned post-processing method that refines boundaries based on edge maps and can in principle be applied on top of many backbones. Since SegFix is orthogonal to our encoder design, we view it as complementary rather than a direct baseline. We have added a reference and short discussion in the related work section and will clarify that SegFix could also be used on top of DUPS predictions, a detailed integration study is left for future work.
>
> ## FPS.
> In the revised version, we will add end-to-end inference FPS for DUPS.
>
> ## Encoder performance on ImageNet-1K.
> DUPS is designed primarily as a segmentation backbone: the encoder is optimized to exploit dense edge maps when learning the upsampling score. In ImageNet-1K pre-training, no such boundary annotations are available, and the upsampling decisions are therefore driven by a random policy. Consequently, the ImageNet-1K accuracy of DUPS should be interpreted as a pre-training proxy rather than as the main indicator of its effectiveness.
>
>
> ## Figures and illustration of the Upsampling block.
> We appreciate the reviewer’s comments on the figures. We have polished Figures 1 and 2 to improve their visual quality and added a visual example showing multi-scale tokens per stage to Figure 1. The internal structure of the Upsampling block is already shown in Figure 2, which is dedicated to this module. Rather than duplicating details, we keep Figure 1 at a higher architectural level and update its caption to explicitly refer to Figure 2 for the Upsampling block design.
>
>
> ## How many tokens are upsampled? Is it a sparse set?
>
> Yes, the upsampling is indeed sparse. At each stage, only a subset of tokens receives high upsampling scores and is split into 2×2 sub-tokens; low-score tokens remain coarse. The average fraction of upsampled tokens is dataset-dependent and varies across stages.
>
> For DUPS-B on ADE20K, the average fractions of currently present tokens upsampled at stages 1–3 are 62\%, 64\%, and 58\%, which correspond to roughly 62\%, 40\%, and 23\% of the total possible tokens at each stage, respectively.
>
> ## Extension to other dense prediction tasks.
> Yes, the method can naturally be extended to other dense prediction tasks where boundary information is available or can be approximated, such as panoptic and instance segmentation. In those settings, one can define edge maps from the instance or panoptic masks and train the upsampling block in the same way as for semantic segmentation. We mention in the conclusion that exploring these extensions is an interesting direction that we leave for future work.

---

### Official Review · Reviewer_emtZ · 2025-10-29

**Soundness:** 2
**Presentation:** 3
**Contribution:** 4
**Rating:** 6
**Confidence:** 4

**Summary:**

This paper proposes DUPS, a coarse-to-fine vision transformer for semantic segmentation, which addresses the inefficiency of uniform and high-to-low resolution architectures by starting with low-resolution tokens and dynamically upsampling only semantically complex regions. Its core innovations include a boundary-aware scoring module for targeted upsampling, a content-adaptive upsampling ratio policy compatible with minibatch training, and an inverted U-Net structure that enables multi-scale token interaction.
Empirically, DUPS demonstrates strong performance across three key benchmarks: it achieves state-of-the-art mIoU on ADE20K and COCO-Stuff with significantly fewer FLOPs, and delivers competitive accuracy on Cityscapes at a fraction of the compute of comparable baselines. Ablation studies further validate the necessity of dynamic upsampling, the inverted U-Net structure, and auxiliary image data, reinforcing the reliability of the proposed method.

**Strengths:**

It proposes a brand-new semantic segmentation architecture combining "dynamic upsampling and coarse-to-fine pipeline", which is different from existing "token pruning" and "fixed scale allocation" methods, showing significant innovation in technical routes.
The efficiency advantage is significant: the FLOPs are lower than those of baseline methods with the same accuracy. Moreover, it maintains competitiveness on real-scene datasets such as Cityscapes, verifying the effectiveness of the method in practical applications.

**Weaknesses:**

Although the paper fully verifies the advantages of the "low-to-high resolution processing pipeline" in computational efficiency (FLOPs), it fails to deeply explain why this mechanism can outperform the traditional "high-to-low resolution processing pipeline" in terms of accuracy (mIoU), resulting in an incomplete demonstration of the performance rationality of its core innovation.
The paper regards the "one-token-one-class" principle as the core logical support for the dynamic upsampling mechanism. However, it fails to provide a clear definition or systematic elaboration of this principle.

**Questions:**

The paper points out that "high-resolution initialization" leads to redundant computation in homogeneous regions. Then, is the dynamic upsampling mechanism starting from low resolution proposed by the authors only intended to reduce computational load? Does this mechanism contribute to improving model performance? Compared with models with high-resolution initialization, what advantages does DUPS have in feature extraction? Since DUPS can reduce redundant computation, can it also demonstrate significant accuracy advantages on test sets with high semantic complexity?

---

> ### Author Response · Authors · 2025-11-21
> **Answer to reviewer emtZ**
>
> We thank the reviewer for the careful reading and the constructive comments. Below we address the main concerns and clarify the design rationale of DUPS.
>
> ## Why DUPS can outperform traditional high-to-low pipelines.
>
> Our mechanism is not only about reducing FLOPs, but also about improving how representational capacity is used. When a token covers multiple semantic classes, its feature vector must encode which classes are present, their relative proportions, and their spatial layout, in addition to position. If, instead, a token predominantly corresponds to a single class, its capacity can be devoted to modeling intra-class variation and higher-order semantics rather than resolving class boundaries, which makes the representation more efficient and expressive for a fixed width.
>
> DUPS is explicitly designed to bias the network toward such class-homogeneous tokens. Concretely, the Upsampling block predicts an upsampling score for each token. If a token is already class-homogeneous, its score is low and it is left unchanged. If a token is likely to mix classes (e.g., it overlaps a semantic boundary), its score is high and it is split into four higher-resolution sub-tokens. The same rule is applied recursively at later stages: whenever a token is predicted to be mixed, it is further split; tokens predicted to be homogeneous are never refined. As a result, mixed regions are repeatedly partitioned into smaller tokens until they become (approximately) single-class, while large homogeneous regions stay coarse. We agree that this rationale was not sufficiently emphasized in the original submission; in the revision we have added a conceptual explanation in the introduction and clarified the mechanism in the method section, replacing the previously vague “one-token-one-class” phrasing with a more precise description of this behavior.
>
>
> ## Is dynamic upsampling only intended to reduce computation? Advantages in feature extraction and accuracy.
>
> Dynamic upsampling from low resolution has two distinct roles. First, the sparse nature of the upsampling is primarily an efficiency mechanism: only a fraction of tokens are ever refined to high resolution, so starting from low resolution and upsampling selectively greatly reduces the overall computational load. Second, and importantly for accuracy, the way we choose which tokens to upsample directly affects feature quality. Because the Upsampling block assigns high scores to tokens that are likely to overlap semantic boundaries and recursively splits only those tokens, high-resolution tokens end up concentrated around boundaries and fine structures, and tend to be more class-homogeneous (as explained above). Their feature dimensions can therefore focus on modeling intra-class variation instead of resolving class mixtures. Mixed-resolution attention then lets these refined tokens exchange information with coarse tokens carrying global context. Compared to a purely high-to-low pipeline that first processes all regions at high resolution and only later compresses them, DUPS spends its high-resolution capacity where it is most semantically useful, which we find leads to stronger features and higher or similar mIoU at lower FLOPs.
>
> The low-to-high part of the architecture is mainly motivated by efficiency: we build global, low-cost features first and only instantiate fine-grained tokens where the scoring module deems them necessary. The subsequent high-to-low leg of the inverted U-Net is added to improve performance: once fine details have been extracted around boundaries, we merge tokens back to a more compact representation with higher channels per token, enabling deeper layers to operate with higher representational capacity at reduced FLOPs.
>
>
> ## Advantages on semantically complex datasets.
> Regarding the question of whether DUPS should show accuracy gains on test sets with high semantic complexity: this is consistent with our empirical results. Datasets such as ADE20K and COCO-Stuff contain many object and stuff categories per image and dense semantic boundaries. On these benchmarks DUPS achieves state-of-the-art or near–state-of-the-art mIoU while using substantially fewer FLOPs than comparable backbones. This is precisely the regime where our semantic-edge aware dynamic upsampling is expected to help most, since many tokens straddle class boundaries and benefit from refinement. On Cityscapes, which is more constrained in terms of scene layout, DUPS remains competitive in accuracy while offering large computational savings.

---

### Meta-Review · Area_Chair_knB2 · 2026-01-04

**Summary:**

This paper proposes the DUPS network, a coarse-to-fine semantic segmentation framework that introduces a boundary-aware scoring and selection module and a content-adaptive per-scale ratio policy. The method achieves competitive segmentation accuracy across multiple datasets. Reviewers raised questions about the novelty of the paper and the clarity of the method description. Although the authors provided explanations in their responses, some reviewers still felt that some parts of method were not well justified. In addition, the presentation is relatively poor. Some experimental results mentioned in the authors’ response also do not appear in the revised manuscript.

**Reviewer Concerns:**

The authors provided responses to the reviewers’ comments. They explained the role of dynamic upsampling mechanism starting from low resolution in terms of efficiency and performance, and clarified its advantages over traditional methods, which addressed most of reviewer emtZ’s concerns. In addition, the experimental section was updated to include fair comparisons with other methods and additional important baselines, which addressed the concerns of reviewers jsQa and 2E8y.
Although the authors elaborated on the differences between the proposed method and existing approaches, reviewers XfA3 and 2E8y stated that their concerns were not fully resolved and the response overstated the novelty.

**Reviewer Scores:**

I believe that none of the four reviewers will change their scores.

---

### Decision · Program_Chairs · 2026-01-26

Reject